# Differences between Clinical Protocols for the Treatment of Coronavirus Disease 2019 (COVID-19) in Andalusia, Spain

**DOI:** 10.3390/ijerph17186845

**Published:** 2020-09-19

**Authors:** Luis M. Pérez-Belmonte, María D. López-Carmona, Juan L. Quevedo-Marín, Michele Ricci, Jesica Martín-Carmona, Jaime Sanz-Cánovas, Almudena López-Sampalo, María D. Martín-Escalante, M. Rosa. Bernal-López, Ricardo Gómez-Huelgas

**Affiliations:** 1Servicio de Medicina Interna, Hospital Regional Universitario de Málaga, Instituto de Investigación Biomédica de Málaga (IBIMA), Universidad de Málaga (UMA), 29010 Málaga, Spain; michele.ricci.sspa@juntadeandalucia.es (M.R.); jessica.martin.sspa@juntadeandalucia.es (J.M.-C.); jaime.sanz.sspa@juntadeandalucia.es (J.S.-C.); almudena.lopez.sampalo.sspa@juntadeandalucia.es (A.L.-S.); rosa.bernal@ibima.eu (M.R.B.-L.); rgh@uma.es (R.G.-H.); 2Centro de Investigación Biomédica en Red Enfermedades Cardiovasculares (CIBERCV), Instituto de Salud Carlos III, 28029 Madrid, Spain; 3Departamento de Bioquímica, Universidad Rey Juan Carlos, 28933 Madrid, Spain; jlquevedo@urj.es; 4Servicio de Medicina Interna, Hospital Costa del Sol, 29603 Marbella, Spain; mariad.m.escalante@juntadeandalucia.es; 5Centro de Investigación Biomédica en Red Fisiopatología de la Obesidad y Nutrición (CIBERobn), Instituto de Salud Carlos III, 28029 Madrid, Spain

**Keywords:** coronavirus disease 2019, management, clinical protocol, differences, antiviral agent, corticosteroids, anakinra, heparin

## Abstract

Our objective was to compare clinical protocols for the treatment of the novel coronavirus disease 2019 (COVID-19) among different hospitals in Andalusia, Spain. We reviewed the current COVID-19 protocols of the 15 largest hospitals in Andalusia. Antiviral treatment, empirical antibacterial agents, adjunctive therapies, anticoagulant treatment, supportive care, hospital organization, and discharge recommendations were analyzed. All protocols included were the latest updates as of July 2020. Hydroxychloroquine in monotherapy was the most frequent antiviral drug recommended for mild respiratory illness with clinical risk factors (33.3%). Combined hydroxychloroquine with azithromycin or lopinavir/ritonavir was found in 40% of protocols. The recommended treatment for patients with mild and moderate pneumonias was different antiviral combinations including hydroxychloroquine plus azithromycin (93.3%) or hydroxychloroquine plus lopinavir/ritonavir (79.9%). Different combinations of hydroxychloroquine and lopinavir/ritonavir (46.7%) and triple therapy with hydroxychloroquine, azithromycin, and lopinavir/ritonavir (40%) were the most recommended treatments for patients with severe pneumonia. There were five corticosteroid regimens, which used dexamethasone, methylprednisolone, or prednisone, with different doses and treatment durations. Anakinra was included in seven protocols with six different regimens. All protocols included prophylactic heparin and therapeutic doses for thromboembolism. Higher prophylactic doses of heparin for high-risk patients and therapeutic doses for patients in critical condition were included in 53.3% and 33.3% of protocols, respectively. This study showed that COVID-19 protocols varied widely in several aspects (antiviral treatment, corticosteroids, anakinra, and anticoagulation for high risk of thrombosis or critical situation). Rigorous randomized clinical trials on the proposed treatments are needed to provide consistent evidence.

## 1. Introduction

The global pandemic of the novel coronavirus disease 2019 (COVID-19) caused by a newly emergent severe acute respiratory syndrome coronavirus 2 (SARS-CoV-2) was firstly recognized in Wuhan (China) in December 2019. It quickly became a global pandemic, spreading worldwide [1]. Spain is one of the most affected countries, with more than 593,000 confirmed cases and more than 29,700 deaths. Andalusia—one of the 17 autonomous communities of Spain and the most populous of them, with a total population of over eight million—has officially registered around 42,640 cases (7.2% of total confirmed cases nationwide) and 1552 deaths (5.2% of total deaths nationwide) [2].

Currently, efficacy has not been fully established for any drug therapy for SARS-CoV-2. Several agents are being used in clinical trials and compassionate-use protocols based on in vitro activity against SARS-CoV-2 or related viruses, and on limited clinical experience [3]. Early detection and optimized supportive care to relieve symptoms and support organ function in more severe presentations are the mainstay of management. Where possible, moderate to severely ill patients should be managed in a hospital setting [4].

There are hundreds of clinical protocols around the world adapted locally to patient characteristics, prevention measures, diagnostic tests, availability of potential therapy options, and possibility of follow-up. The use of clinical protocols in health care aims to provide practitioners with locally agreed information about what is currently the recommended approach for a specific practice. Such protocols should be systematically developed and based on an evaluation of the current best evidence. Thus, they also have the potential advantage of reducing unnecessary variations in care and contributing to evidence-based health care [5]. Based on this premise, we conducted this study, whose main objective was to compare updated clinical protocols for treatment of COVID-19 among the largest university hospitals in Andalusia, Spain. We hypothesized that clinical protocols could vary in several aspects among Andalusian Hospitals, regardless of whether they belong to the same local health system administration and have similar resources.

## 2. Materials and Methods

In this study, we used the clinical information provided in the COVID-19 protocols of the largest university hospitals in Andalusia. In Andalusia, hospitals are stratified into 4 categories according to service area and number of medical and surgical specialties [6]: (1) regional university hospitals (referral hospital in the whole autonomous community with all specialties available); (2) specialized university hospitals (referral hospital in a province and with wide number of specialties available); (3) basic general hospitals (referral hospital in a region and with all basic specialties available); (4) highly-specialized hospitals (referral hospital in a local area and with all basic specialties available). We included 6 regional university hospitals and 9 specialized university hospitals: Hospital Universitario Reina Sofía (Córdoba), Hospital Universitario Juan Ramón Jiménez (Huelva), Hospital Universitario Virgen del Rocío (Sevilla), Hospital Universitario Virgen Macarena (Sevilla), Hospital Virgen de Valme (Sevilla), Complejo Hospitalario Torrecárdenas (Almería), Hospital Universitario de Jerez de la Frontera (Jerez de la Frontera), Hospital Universitario Puerta del Mar (Cádiz), Hospital de Puerto Real (Puerto Real), Hospital Universitario Virgen de las Nieves (Granada), Hospital Universitario San Cecilio (Granada), Complejo Hospitalario de Jaén (Jaén), Hospital Universitario Virgen de la Victoria (Málaga), Hospital Costa del Sol (Marbella), and Hospital Regional Universitario de Málaga (Málaga). This study was approved by the Institutional Research Ethics Committee of Málaga on 1 May 2020 (Ethical code: PI-Prot-0520)

We excluded 19 basic general hospitals and 15 highly-specialized hospitals. Normally, the treatment recommendations adopted in the reference hospital indicate how their area of influence proceeds.

Antiviral treatment, empirical antibacterial agents, adjunctive therapies, anticoagulant treatment, supportive care (oxygen, intravenous fluids, monitoring, high-flow nasal oxygen/noninvasive ventilation, and mechanical ventilation, among others), dosage, method of administration and duration of treatment, drug side effects and interactions, nonrecommended treatment or recommendation for other drugs, hospital circuits, ward organization and care planning, and discharge recommendations were collected from all clinical protocols for COVID-19.

COVID-19 protocols contained patient groups according to the patient’s clinical condition: mild upper respiratory illness with and without clinical risk factors (aged ≥60 years old and comorbidities), mild pneumonia (defined as CURB-65 Severity Score [7] ≤1 or Pneumonia Severity Index [8] I or II, basal oxygen saturation >95%, respiration rate at rest <22, unilateral opacity, D-dimer <1000 ng/mL (normal range: 220–500 ng/mL), ferritin <1000 mcg/L (normal range: 8–252 mcg/L), interleukin-6 <40 pg/mL (normal range: <4.4 pg/mL)), moderate pneumonia (defined as CURB-65 Severity Score ≥2 or Pneumonia Severity Index ≥III, basal oxygen saturation >92–95%, respiration rate at rest 22–30, bilateral opacities, D-dimer >1000 ng/mL, ferritin >1000 mcg/L, interleukin-6 <40 pg/mL), and severe pneumonia (defined as sepsis or shock with acute respiratory distress syndrome (ARDS), basal oxygen saturation <92%, respiration rate at rest >30, bilateral opacities, D-dimer >3000 ng/mL, ferritin >1000 mcg/L, interleukin-6 >40 pg/mL, elevated troponin I). Prophylaxis in suspected cases was not taken into consideration.

All protocols included were obtained from COVID-19 teams of each hospital and were the latest updates as of July 2020. Protocols are only available for healthcare providers on previous request to COVID-19 teams of each hospital. The update of protocols was routinely made by COVID-19 teams when they considered necessary.

All variables analyzed were categorical and are shown as the absolute value and percentage, respectively. Statistical analyses were performed using SPSS Statistics for Windows, version 15.0(IBM, Armonk, NY, USA).

## 3. Results

Symptomatic treatment, recommendations for infection prevention and control, and recommendations for early identification of COVID-19 alarm symptoms were adopted for patients with mild upper respiratory illness without clinical risk factors and confirmed cases without symptoms.

Antiviral treatment was recommended for patients with mild upper respiratory illness with clinical risk factors and patients with mild, moderate, and severe pneumonia. Patients could be treated with monotherapy using hydroxychloroquine or lopinavir/ritonavir; dual therapy with hydroxychloroquine plus azithromycin or lopinavir/ritonavir; or triple therapy with hydroxychloroquine, azithromycin, and lopinavir/ritonavir. Hydroxychloroquine could be used in four different regimens, and lopinavir/ritonavir could be used in two different regimens, with varied dosages and duration of treatment—either a short (5-day) regimen or long (10-day) regimen. Azithromycin was recommended at the same dosage and duration of treatment. Protocols included the combination of all four hydroxychloroquine regimens with azithromycin. Regimens 1 and 3 of hydroxychloroquine were combined in dual therapy with the two regimens of lopinavir/ritonavir, which in turn were combined with azithromycin in triple therapy. Chloroquine 500 mg twice per day was recommended as an alternative when hydroxychloroquine was not available. In patients with three antiviral agents, lopinavir/ritonavir could be replaced with once-daily remdesivir if the patient was enrolled in a clinical trial. The dosage and durations of antiviral treatments are shown in Table 1.

For treating mild respiratory illness in patients with clinical risk factors, up to six antiviral options were found. Monotherapy with regimen 1 of hydroxychloroquine was the most frequent antiviral treatment recommended (5/15 protocols, 33.3%), followed by hydroxychloroquine regimen 1 plus azithromycin (3/15, 20%). Dual antiviral treatment using hydroxychloroquine with azithromycin or lopinavir/ritonavir in combination was found in 40% of protocols. No treatment was recommended in two protocols. In regard to mild pneumonia, 14 protocols recommended dual therapy: seven recommended hydroxychloroquine plus azithromycin, and seven recommended hydroxychloroquine plus lopinavir/ritonavir. Only one protocol did not recommend any antiviral treatment. For moderate pneumonia, hydroxychloroquine regimen 1 in combination with lopinavir/ritonavir regimen 2 was the most recommended treatment (5/15, 33.3%). Different combinations of hydroxychloroquine and lopinavir/ritonavir (7/15, 46.7%) and triple therapy with hydroxychloroquine regimen 3, and azithromycin and lopinavir/ritonavir regimen 2 (6/15, 40%) were the most recommended treatments for patients with severe pneumonia. Most protocols showed a preference for dual or triple antiviral therapy for patients with pneumonia. Further details about recommended antiviral treatment of COVID-19 are summarized in Table 2.

All protocols recommended initiating empirical antibacterial treatment when secondary bacterial infection was suspected, according to local antibacterial therapy guidelines. Empirical antibiotics were also indicated when pneumonia was diagnosed in seven protocols regardless of suspected bacterial infection. Amoxicillin plus clavulanic acid or ceftriaxone were the main antibiotics recommended for use.

All protocol considered the management of ARDS with signs of cytokine release syndrome, which included elevation of interleukin-6, fibrinogen, D-dimer, and C-reactive protein levels [9]. The therapies (corticosteroids, anticytokine or immunomodulatory agents, and immunoglobulin therapy) were recommended once these syndromes were observed. The choice of which therapy to use was made by physicians according to their own clinical judgment. Corticosteroids were widely included in all protocols; indeed, only 2 out of 15 hospitals did not include any corticosteroid regimen. There was a total of five corticosteroid regimens, which used dexamethasone, methylprednisolone, or prednisone. Dexamethasone was recommended in eight protocols with two different regimens: 20 mg (5/8, 62.5%) or 40 mg (3/8, 37.5%) daily. Methylprednisolone was included in 13 protocols with 2 different regimens: 125–250 mg (8/13, 61.5%) or 1 mg/kg (5/13, 38.5%) daily. Only one protocol included prednisone with a single regimen of 40 mg daily. The duration of treatment varied from 3 to 10 days, depending on the patient’s clinical condition. Tocilizumab was recommended in all protocols with the same dosage (400 or 600 mg) according to body weight. Anakinra was included in seven protocols with six different regimens. Immunoglobulins were included as possible therapy in three protocols. In six protocols, other optional therapies were included, such as sarilumab, baricitinib, colchicine, ciclosporin A, sirolimus, tacrolimus, and vitamin D, according to clinical trials. Dosage and durations of corticosteroids, anticytokine or immunomodulatory agents, and immunoglobulin therapy are shown in Table 3.

All protocols included anticoagulant treatment (low-molecular-weight heparin as the first option and unfractionated heparin as an alternative) and recommended prophylactic heparin during hospitalization. In 8 of the 15 COVID-19 protocols (53.3%), the dose of prophylactic low-molecular-weight heparin was increased if there was a high risk of thrombosis (severe COVID-19 with evidence of cytokine release syndrome, previous venous thromboembolism or acute ischemic artery disease, or D-dimer >3000 ng/mL). In five protocols (33.3%), heparin was increased to therapeutic doses if the condition was critical or if there was a progressive increase in D-dimer levels. All protocols included therapeutic heparin if there was evidence of venous thromboembolism. At discharge, prophylactic heparin was recommended if D-dimer >1500-3000 ng/dL for 7 days or during the time of the expected severe immobilization in seven protocols (46.6%). Adjustments of heparin doses were indicated in the respective protocols.

Supportive care, hospital circuits, ward organization, and care planning were included in all protocols. Drug side effects (including follow-up electrocardiogram for QT prolongation), interactions, nonrecommended treatment, or recommendation for other drugs were included in 11 protocols (73,3%), and discharge recommendations in 12 (80%).

## 4. Discussion

This study showed that the COVID-19 protocols of the largest hospitals in Andalusia (Spain) varied widely in several aspects, including antiviral treatment: they used different dosages, durations, and combinations of treatment for the same clinical condition. Corticosteroid and anakinra regimens varied among hospitals, whereas tocilizumab was uniform across all protocols. Although anticoagulant treatment was recommended in all protocols, including prophylactic heparin during hospitalization for all patients and therapeutic heparin for patients with evidence of venous thromboembolism, only a few protocols included higher prophylactic doses for patients at high risk of thrombosis and therapeutic doses for patients in critical condition or who experience a progressive increase in D-dimer levels. On the other hand, all treatment protocols included the same patient groups, defined according to the clinical situation, symptomatic treatment and recommendations for infection prevention and control, supportive care and hospital organization, equivalent empirical antibacterial agents, drug warnings and interactions, and discharge recommendations.

The clinical presentation of the novel COVID-19 may vary from mild cases with fever, fatigue, and cough to moderate–severe cases involving pneumonia and multiorgan failure [10], and several factors may predispose COVID-19 patients to adverse outcomes. The older age, critical disease, and high levels of inflammatory markers have been associated with increased risk of death [11]. Patients with moderate to severe forms of COVID-19 normally require hospitalizations and pharmacological treatments. However, currently, there is no solid evidence that any potential drug improves outcomes in patients with suspected or confirmed COVID-19 [12,13]. Only clinical experience and treatment guidance based on repurposed and experimental treatments are available.

Several antiviral agents have been proposed for use in combatting COVID-19 based on apparent in vitro activity [3]. One of the most widely used agents has been hydroxychloroquine and chloroquine. Although studies from China [14] and France [15] have reported improved radiologic findings, improved viral clearance, and reduced disease progression with the use of chloroquine and hydroxychloroquine compared to standard supportive care, their efficacy remains inconclusive, and further studies are warranted. In addition, it seems that hydroxychloroquine may act synergistically in combination with azithromycin [3,15]. Recently, in a large observational study from Italy, hydroxychloroquine was associated with a 30% lower risk of death in COVID-19 hospitalized patients [16]. Despite these results, these studies had severe methodological limitations, such as lack of randomization, lack of covariate-adjusted analysis, and potential selection bias. RECOVERY trial—the largest randomized controlled study on hydroxychloroquine—has suggested that hydroxychloroquine might not reduce deaths and might increase length of hospital stay [17]. Chloroquine dosage has been 500 mg orally once or twice daily. However, hydroxychloroquine dosage recommendations have varied from a total daily dose of 400 mg (with a loading dose of 400 mg twice daily for 1 day) to 600 mg orally, based on safety and clinical experience for other diseases [3]. With this limited evidence, international guidelines with consensus statements on the treatment of COVID-19 have not included any recommendation about using hydroxychloroquine/chloroquine as a potential treatment [4,18,19,20], but rather have only suggested its use in the context of a clinical trial [21,22]. In our study, monotherapy with 400 mg hydroxychloroquine twice daily the first day followed by 200 mg twice daily from day 2 to day 5, alone and in combination with azithromycin, were the preferred recommended regimens among protocols in Andalusia for treating mild respiratory illness with clinical risk factors. For mild pneumonia, dual therapies consisting of hydroxychloroquine-azithromycin or hydroxychloroquine-lopinavir/ritonavir were the most frequent treatments recommended, with regimens that included different dosages and durations of treatment. For moderate pneumonia, a shorter regimen of hydroxychloroquine in combination with a longer regimen of lopinavir/ritonavir was the most recommended treatment among all protocols. For the most severe form of pneumonia, the triple therapy with a longer regimen of hydroxychloroquine and lopinavir/ritonavir in combination with azithromycin was the most commonly indicated treatment. Thus, more complex and longer antiviral therapies were recommended according to the severity of COVID-19 in our protocols.

Lopinavir/ritonavir has had the same limitations of use as hydroxychloroquine/chloroquine, with no clear benefits beyond standard care [3,23]. Considering the uncertainty and the likely increase in gastrointestinal side effects, principal guidelines around the world have not contemplated its use in patients with COVID-19 [4,18,19,20] or have only considered it in the context of a clinical trial [21,22]. Remdesivir was only considered when patients were enrolled in a clinical trial. Randomized controlled trials have reported that remdesivir reduces the duration of mechanical ventilation and time to symptom resolution, but its effect on mortality and other adverse outcomes remains uncertain [24]. Other antivirals were not routinely included in our protocols. Antiviral agents such as oseltamivir, umifenovir, favipiravir, or ribavirin as well as miscellaneous agents such as interferon-α or interferon-β are being studied for use as possible treatments for COVID-19 [3]. Currently, principal guidelines have either not reported any recommendation for their use [4,18,19,20,21] or have suggested not using them in patients with COVID-19 [22].

In regard to management of ARDS with evidence of cytokine release syndrome, several adjunctive therapies have been proposed. The rationale for the use of corticosteroids is to reduce host inflammatory responses in the lungs. The potential harm, such as delayed viral clearance, increased risk of secondary infection, and hyperglycemia episodes, have led to caution in their routine use in COVID-19 patients unless the patient had other conditions for which these are indicated [3,4,18,19,20,21]. However, in observational and randomized studies focused on hospitalized patients with COVID-19 pneumonia, the administration of corticosteroids reduced the risk of mortality [25,26,27]. Based on this fact, a recently published guideline suggested using 40 mg methylprednisolone intravenously for 10 days in patients with severe COVID-19 and ARDS [22]. In our study, corticosteroids were included in 87.7% of the protocols. Methylprednisolone was the most frequently included corticosteroid in the protocols (13/15 protocols) with high-dose regimens (≥1 mg/kg/day) and a variable duration of treatment (between 3 and 10 days). High-dose dexamethasone (20 or 40 mg) was included in 53.3% of the protocols and prednisone in just one protocol (40 mg per day for 5 days). Monoclonal antibodies targeting key inflammatory citokines or other aspects of the innate immune response are another potential treatment for COVID-19 [28]. Tocilizumab, an anti-interleukin 6 receptor antibody, has been used in small studies with early reports of success. A dose of 400 mg was associated with clinical improvements and successful discharge, with most patients only receiving a single dose [29,30]. Several randomized clinical trials on tocilizumab, either alone or in combination, in patients with COVID-19 with severe pneumonia are underway, and its use was included in the Chinese national treatment guidelines [30,31]. Likewise, tocilizumab was recommended in all protocols in Andalusian hospitals; the protocols indicate a single dose of 400 mg or 600 mg, according to body weight, and a second dose 12-24 h later, if required. Although tocilizumab is a promising therapy, the data currently available are still too limited to draw any conclusion about its viability. Anakinra, an interleukin 1 receptor antagonist, has also been proposed in order to reduce hyperinflammation and respiratory distress in patients with SARS-CoV-2 infection, but here, too, limited evidence has been published [32]. In this study, less than half of the protocols included anakinra in their recommendations, and six different regimens were described with significant differences in dosage (ranging from 100 to 400 mg per day) and duration of treatment (between 3 and 10 days). Other monoclonal antibodies, immunomodulatory agents, and immunoglobulin therapy are in clinical trials for the treatment of COVID-19-associated cytokine release syndrome [3].

As recent studies have found, severe COVID-19 is commonly complicated with coagulopathy, including disseminated intravascular coagulation and venous thromboembolism [33,34]. For this reason, the administration of heparin has been recommended for COVID-19 patients according to expert consensus [4,31], although its efficacy remains to be validated. In a study of 449 patients with severe COVID-19 and sepsis-induced coagulopathy criteria or with markedly elevated D-dimer levels (higher than sixfold of upper limit of normal), anticoagulant therapy, mainly with low-molecular-weight heparin, was associated with better prognosis [34]. Beyond severe coagulopathy, special attention to venous thromboembolism prophylaxis is necessary in the management of COVID-19 [35]. Low-molecular-weight heparin is preferred over unfractionated heparin in order to reduce patient contact (depending on the patient’s bleeding risk and creatinine clearance) [36]. We found that all protocols analyzed included anticoagulant treatment, with low-molecular-weight heparin as the first option. It was recommended that all hospitalized COVID-19 patients receive prophylactic heparin. This recommendation was extended to after discharge if necessitated by the patient’s clinical condition in 46.6% of protocols. An increase in prophylactic heparin doses was recommended in 53.3% of the protocols for patients at high risk of thrombosis. In five protocols (33.3%), heparin was increased to therapeutic doses if the condition was critical or if there was a progressive increase in D-dimer levels. All protocol included therapeutic heparin if there was evidence of venous thromboembolism.

Our findings show significant differences among local protocols throughout Andalusia (Spain) regarding the management of COVID-19 patients. There are some possible explanations for these disparities, including a lack of strong evidence about medical therapies that have been definitively reported to be effective at this time, the local availability of therapies, and administrative decisions about organization and care planning. Although consensus statements in some international guidelines recommend against routine use of most antiviral agents, the eagerness to help our patients leads healthcare providers to explore different therapeutic strategies, balancing benefits and detriments. The same issue occurs with adjunctive therapies that are only suggested for treating ARDS with evidence of cytokine release syndrome as well as with anticoagulation in patients at high risk of thrombosis. There is greater consensus on the routine use of thromboembolism prophylaxis or anticoagulation for established venous thromboembolism. All protocols analyzed, in accordance with other guidelines, emphasize the prompt implementation of recommended infection prevention and supportive care for complications.

These results are important because they show significant differences among the recommended treatments for COVID-19, according to the treatment protocols of large university hospitals in a region of Spain. They reveal a need for specifically designed randomized clinical trials to determine the most appropriate evidence-based treatment regimen. COVID-19 is having a strong impact in several countries, including Spain, in several aspects, especially health, economic, and social. There is a quickly growing body of evidence on this topic trying to find the best practice for the treatment of symptomatic patients with COVID-19 [37].

This study has several limitations. First, we only analyzed the data provided in the hospitals’ treatment protocols, and no information on treated actually provided to patients was obtained. Furthermore, we did not know the degree of protocol implementation in the hospital or its coverage area. Second, only the protocols of the largest hospitals of Andalusia were analyzed in this study, even though there are many smaller hospitals and healthcare centers. Normally, the treatment recommendations adopted in reference hospitals indicate how their area of influence proceeds, but this cannot be definitively assured. Third, our study focused only on adult patients, and the recommendations were not applicable to pediatric populations. Finally, the recommendations provided in the protocols are based on local expert opinions, clinical experience in managing COVID-19 patients, and current international evidence regarding major proposed treatments, whether repurposed or experimental, for COVID-19. Therefore, no strong therapeutic recommendations could be drawn from the protocols analyzed in this study.

## 5. Conclusions

In conclusion, this study showed that the COVID-19 protocols of the largest hospitals in Andalusia, Spain, varied widely in several aspects, including antiviral treatment; corticosteroids; anakinra; prophylactic heparin for patients at high risk of thrombosis; therapeutic doses of heparin for patients in critical condition or with a progressive increase in D-dimer levels; and different dosage, duration, and combinations of treatment for patients in the same clinical condition. The use of tocilizumab for selected patients, prophylactic heparin for all patients, therapeutic heparin for evidence of venous thromboembolism, general symptomatic and supportive care, and hospital organization were recommended in all protocols. Rigorous randomized clinical trials on the proposed interventions are needed in order to provide solid evidence.

## Figures and Tables

**Table 1 ijerph-17-06845-t001:** Dosage and durations of antiviral treatment for coronavirus disease 2019.

**Hydroxychloroquine**	
Regimen 1	400 mg every 12 h the first day followed by 200 mg every 12 h for 4 days
Regimen 2	400 mg every 12 h the first day followed by 200 mg every 8 h for 4 days
Regimen 3	400 mg every 12 h the first day followed by 200 mg every 12 h for 6–9 days
Regimen 4	400 mg every 12 h the first day followed by 200 mg every 8 h for 6–9 days
**Lopinavir/ritonavir**	
Regimen 1	200/50 mg every 12 h for 5 days
Regimen 2	200/50 mg every 12 h for 7–10 days
**Azithromycin**	
Regimen	500 mg every 24 h the first day followed by 250 mg every 24 h for 4 days
**Remdesivir**	
Regimen	200 mg every 24 h the first day followed by 100 mg every 24 h for 9 days

Note: mg: milligram.

**Table 2 ijerph-17-06845-t002:** Antiviral treatment protocols for coronavirus disease 2019 according to clinical condition.

Antiviral Treatment	Mild Respiratory Illness with Clinical Risk Factors	Mild Pneumonia	Moderate Pneumonia	Severe Pneumonia
No	2 (13.3%)	1 (6.7%)	NR	NR
H1	5 (33.3%)	NR	NR	NR
H2	1 (6.7%)	NR	NR	NR
LR1	1 (6.7%)	NR	NR	NR
H1 + A	3 (20.0%)	4 (26.6%)	1 (6.7%)	NR
H2 + A	1 (6.7%)	2 (13.3%)	1 (6.7%)	NR
H3 + A	NR	1 (6.7%)	2 (13.3%)	1 (6.7%)
H4 + A	NR	NR	1 (6.7%)	1 (6.7%)
H1 + LR1	2 (13.3%)	3 (20.0%)	NR	NR
H1 + LR2	NR	1 (6.7%)	5 (33.3%)	3 (20.0%)
H3 + LR1	NR	NR	1 (6.7%)	NR
H3 + LR2	NR	3 (20.0%)	1 (6.7%)	4 (26.6%)
H1 + A + LR1	NR	NR	1 (6.7%)	NR
H1 + A + LR2	NR	NR	1 (6.7%)	NR
H3 + A + LR1	NR	NR	1 (6.7%)	NR
H3 + A + LR2	NR	NR	NR	6 (40.0%)

H1: hydroxychloroquine 400 mg/12 h (day 1) + 200 mg/12 h (day 2 to 5) (regimen 1); H2: hydroxychloroquine 400 mg/12 h (day 1) + 200 mg/8 h (day 2 to 5) (regimen 2); H3: hydroxychloroquine 400 mg/12 h (day 1) + 200 mg/12 h (day 2 to 7–10) (regimen 3); H4: hydroxychloroquine 400 mg/12 h (day 1) + 200 mg/8 h (day 2 to 7–10) (regimen 4); A: Azithromycin 500 mg/24 h (day 1) + 250 mg/24 h (day 2 to 5); LR1: lopinavir/ritonavir 200/50 mg/12 h (day 1 to 5) (regimen 1); LR2: lopinavir/ritonavir 200/50 mg/12 h (day 1 to 7–10) (regimen 2); NR: not reported.

**Table 3 ijerph-17-06845-t003:** Dosage and duration of treatment of corticosteroids, anticytokine or immunomodulatory agents, and immunoglobulin therapy for coronavirus disease 2019.

**Corticosteroids**	
**Dexamethasone**	
Regimen 1	20 mg every 24 h for 3–5 days
Regimen 2	40 mg every 24 h for 3–5 days
**Methylprednisolone**	
Regimen 1	125–250 mg every 24 h for 3–10 days
Regimen 2	1 mg/kg every 24 h for 3–10 days
**Prednisone**	40 mg every 24 h for 5 days
Anticytokine or immunomodulatory agents	
**Tocilizumab**	
Regimen 1 (Body weight <75 kg)	Single dose of 400 mg Second dose of 400 mg could be administered 12–24 h later if required
Regimen 2 (Body weight ≥75 kg)	Single dose of 600 mg. Second dose of 400 mg could be administered 12–24 h later if required
**Anakinra**	
Regimen 1	100 mg every 24 h for 3 days
Regimen 2	200 mg every 12 h for 3 days
Regimen 3	100 mg every 6 h (day 1), 100 mg every 12 h (day 2) Optionally 100 mg every 12 h (day 3)
Regimen 4	200 mg every 24 h (day 1) and 100 mg every 24 h (day 2–5)
Regimen 5	100 mg every 6 h (day 1), 100 mg every 8 h (day 2), and 100 mg every 12 h (day 3–7)
Regimen 6	200 mg every 24 h (day 1), and 100 mg every 6-8 h for 5-10 days
**Immunoglobulins**	
Regimen	400 mg/kg/day for 5 days

kg: kilogram; mg: milligram.

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
