# Peer review of "Differences between Clinical Protocols for the Treatment of Coronavirus Disease 2019 (COVID-19) in Andalusia, Spain"

_ijerph, 2020, doi:10.3390/ijerph17186845_

Round 1

Reviewer 1 Report

Comments and Suggestion

The manuscript entitled “Differences between clinical protocols for the management of coronavirus disease 2019(COVID-19) in Andalusia, Spain.”, presented by Luis M. Perez-Belmonte et al., deal with discrepancies among clinical protocols in 15 Andalusian hospital, points out that protocols varied in several aspects, especially when it comes to a drug dosing.

Clinical protocols are formal pathways, which should provide standardized algorithms for caring for patients with specific condition. They are used to help implement evidence-based data and especially reduce unnecessary practice variation in many medical fields.

Clinical protocols in  health environment in such a busy time are immensely useful. Although the pandemic continuous, still clinicians have a little evidence of having effective   treatment of  COVID-19. Thus any provided information which help gain more knowledge in this topic is utmost needed. This reviewed manuscript is suitable for publication, however some important points need clarification before.

Major

Introduction

In this paragraph, the short explanation - what for are the clinical protocols, why it could be useful (or opposite) for health care providers, is missing.

Materials and methods

Line 78 –  re-phrased into - We included  n- regional university hospitals and n-specialized university hospitals.

There is no information how the protocols were obtained, via hospital website?, through a special survey? From medical director?

How many hospitals was considered, was any excluded, what was the reason to chose only those 15?Are the rest of the hospital in Andalusia area lack of the protocols? (yes, some explanations is in limitation paragraph, it would be good to move and explain in this paragraph)

Was the protocol on-line available? access was free for every healthcare provider or only available for dedicated department?

Line 92-93   Data was collected at this point of time? How often was the protocol updated at the hospital site, every week? months or once written was not updated?

Results

Line 98 - Mild respiratory illness – what exactly is behind this description?

Table 4. Is not necessary, those information could be obtain from the drug leaflet 

Discussion

Line 331-332 I could not agree with this opinion. Clinical protocol is already for the healthcare provides a recommendation what it could be done in specific medical situation. An analysis done by the authors, in my opinion, was not meant to give strong therapeutic recommendations, but opposite, find the differences and conclude what was the reason of phenomenon.

Suggestion for authors: Many other data obtained from the protocol according to “ Materials and Methods” paragraph  (line 87-93), however, only the descriptions of differences in treatment regimens is discussed broadly in paragraph 3,  in contrast readers are given short description of rest of the variables (line 189-192). As the authors focused on medication mostly, it would be could idea to stick to comparison medications regime only. This in turns, would influence on the title – instead of management – treatment regimens, change accordingly abstract and materials and methods 

Minor

Abstract - informative, could be added the number of  collected protocols

Keywords –add word ‘ comparison’ or ‘differences’

Intruduction

Line 50  not necessary capital letter in ‘Coronavirus’ word

Line 54   extra s after communities

Line 56 update numbers and  reference 2

Line 60- “more severe illnesses” – re-phrase, as it is not clear here what kind illnesses?

Materials and methods

Line 71- is that really observational study?

Line 74- typos ‘hospita’

Line 87-91 –add patient`s clinical condition since the category of severity of pneumonia is describe in results

Results

Line 97-107 CURB-65 Severity Score <1 should  be less and equal; the norm of mentioned lab test are not so obvious or a bit different from laboratory to laboratory, consider to add normal range

Line 112-115- patient had the possibility of being treated – re-phrase

Line 152- add reference

Line 157 –add reference (e.g Teijaro JR. Cytokine storms in infectious diseases. Semin Immunopathol. 2017;39(5):501-503. doi:10.1007/s00281-017-0640-2)

Line 156-157 was those lab test mandatory according to clinical protocols? It is not clear indication for immunomodulatory agents to be used, which were the first choice - steroids? other  immunomodulatory medication

Line 186 –‘severe’ or prolonged?

Discussion

Line 210- ‘and respiratory and kidney failure ’ change to multiorgan failure

Line 239- the severest – the most severe

Line 240-241 – ‘Thus, more complex and longer antiviral therapies were recommended according to the severity of COVID-19 in our protocols.’ – what the information authors would like to pass to the readers? Knowing that there is no solid evidence for any therapy to be true effective, about what,  authors informed us couple of times through this manuscript

Line 285 – what kind of type one-off recommendation of adjunctive therapies authors are mentioned?

Line 289-303 – is repetition of what is described in paragraph 3

References

Review references and upgrade as most of them were published in April ‘20

Tables

Table 2.

In last column – severe pneumonia - sum up numbers give 14 not 15, what regime is missing?

Table 1 and 3 – use bold to underline agents

Author Response

Lyn Zhao

Assitant Editor

International Journal of Environment Research and Public Health

September 14, 2020

Dear Dr. Lyn Zhao:

Thank you for reviewing and considering our manuscript, “Differences between clinical protocols for the management of coronavirus disease 2019 (COVID-19) in Andalusia, Spain” for possible publication in the International Journal of Environment Research and Public Health. We greatly appreciate the comments provided to us by the reviewers.

We have made revisions to our manuscript, according the recommendations of the reviewers, and are now resubmitting it to you for your consideration. The reviewers’ comments and our corresponding responses are listed below.

Thank you very much for your kind attention on this matter.

Sincerely

Corresponding authors on behalf of all co-authors.

Luis M. Pérez-Belmonte, MD, PhD

María D. López-Carmona, MD, PhD

luismiguelpb1984@gmail.com; mdlcorreo@gmail.com

Responses to Reviewers’ Comments:

Reviewer #1:

The manuscript entitled “Differences between clinical protocols for the management of coronavirus disease 2019(COVID-19) in Andalusia, Spain.”, presented by Luis M. Perez-Belmonte et al., deal with discrepancies among clinical protocols in 15 Andalusian hospital, points out that protocols varied in several aspects, especially when it comes to a drug dosing.

Clinical protocols are formal pathways, which should provide standardized algorithms for caring for patients with specific condition. They are used to help implement evidence-based data and especially reduce unnecessary practice variation in many medical fields.

Clinical protocols in health environment in such a busy time are immensely useful. Although the pandemic continuous, still clinicians have a little evidence of having effective   treatment of COVID-19. Thus any provided information which help gain more knowledge in this topic is utmost needed. This reviewed manuscript is suitable for publication, however some important points need clarification before.

Major

Introduction

In this paragraph, the short explanation - what for are the clinical protocols, why it could be useful (or opposite) for health care providers, is missing.

Authors’ reply: We really appreciate this comment in order to improve our paper. We have added a short explanation about this issue (“The use of clinical protocols in health care aims to provide practitioners with locally agreed information about what is currently the recommended approach for a specific practice. Such protocols should be systematically developed and based on an evaluation of the current best evidence. They, thus also have the potential advantage of reducing unnecessary variations in care and contributing to evidence-based health care [5]”) (line 65 to 69). We have adapted the references after this change.

Materials and methods

Line 78 –  re-phrased into - We included  n- regional university hospitals and n-specialized university hospitals.

Authors’ reply: We really appreciate this comment in order to improve our paper. We have re-phrased it (“We included 6 regional university hospitals and 9 specialized university hospitals”) (line 82).

There is no information how the protocols were obtained, via hospital website?, through a special survey? From medical director?

Authors’ reply: We really appreciate this comment in order to improve our paper. We have added this information (“All protocols included were obtained from COVID-19 teams of each hospital”) (line 100).

How many hospitals was considered, was any excluded, what was the reason to chose only those 15? Are the rest of the hospital in Andalusia area lack of the protocols? (yes, some explanations is in limitation paragraph, it would be good to move and explain in this paragraph).

Authors’ reply: We really appreciate this comment in order to improve our paper. We only considered the largest hospitals of Andalusia in this study. Basic general hospitals and highly-specialized hospitals were not included in the analysis because normally the treatment recommendations adopted in reference hospital indicate how their area of influence proceeds. We have included this information in Material and Methods (line 91-93)

Was the protocol on-line available? access was free for every healthcare provider or only available for dedicated department?

Authors’ reply: We really appreciate this comment in order to improve our paper. Protocols are only available for healthcare providers on previous request. We have added this information in Materials and Methods (line 101-102).

Line 92-93   Data was collected at this point of time? How often was the protocol updated at the hospital site, every week? months or once written was not updated?

Authors’ reply: We really appreciate this comment in order to improve our paper. Data was collected at this point of time. All protocols included were the latest updates as of July 2020. The update of protocols was routinely made by COVID-19 teams when they considered necessary. We have added this information in Materials and Methods (line 102-103).

Results

Line 98 - Mild respiratory illness – what exactly is behind this description?

Authors’ reply: We really appreciate this comment in order to improve our paper. We mean “mild upper respiratory illness”. We have added it (line 108).

Table 4. Is not necessary, those information could be obtain from the drug leaflet.

Authors’ reply: We really appreciate this comment in order to improve our paper. We have deleted this table and its reference in the text.

Discussion

Line 331-332 I could not agree with this opinion. Clinical protocol is already for the healthcare provides a recommendation what it could be done in specific medical situation. An analysis done by the authors, in my opinion, was not meant to give strong therapeutic recommendations, but opposite, find the differences and conclude what was the reason of phenomenon.

Authors’ reply: We really appreciate this comment in order to improve our paper. We totally agree with this comment.

Suggestion for authors: Many other data obtained from the protocol according to “ Materials and Methods” paragraph  (line 87-93), however, only the descriptions of differences in treatment regimens is discussed broadly in paragraph 3,  in contrast readers are given short description of rest of the variables (line 189-192). As the authors focused on medication mostly, it would be could idea to stick to comparison medications regime only. This in turns, would influence on the title – instead of management – treatment regimens, change accordingly abstract and materials and methods.

Authors’ reply: We really appreciate this comment in order to improve our paper. We have changed accordingly this recommendation the title (line 3), abstract (line 24), and Materials and Methods (line 94).

Minor

Abstract - informative, could be added the number of collected protocols

Authors’ reply: We really appreciate this comment in order to improve our paper. We have added this information in the abstract (line 26).

Keywordsadd word ‘comparison’ or ‘differences’

Authors’ reply: We really appreciate this comment in order to improve our paper. We have added this information in the keywords (line 26).

Introduction

Line 50  not necessary capital letter in ‘Coronavirus’ word

Authors’ reply: We really appreciate this comment in order to improve our paper. We have delated it.

Line 54   extra s after communities

Authors’ reply: We really appreciate this comment in order to improve our paper. We have delated it.

Line 56 update numbers and reference 2

Authors’ reply: We really appreciate this comment in order to improve our paper. We have updated numbers and reference 2. 

Line 60- “more severe illnesses” – re-phrase, as it is not clear here what kind illnesses?

Authors’ reply: We really appreciate this comment in order to improve our paper. We have re-phrased it. Instead of illnesses, we have written “presentations”. 

Materials and methods

Line 71- is that really observational study?

Authors’ reply: We really appreciate this comment in order to improve our paper. We have delated the word “observational”.

Line 74- typos ‘hospita’

Authors’ reply: We really appreciate this comment in order to improve our paper. We have added the “s”. 

Line 87-91 –add patient`s clinical condition since the category of severity of pneumonia is describe in results.

Authors’ reply: We really appreciate this comment in order to improve our paper. We have added the patient’s clinical condition in the Materials and Methods.

Results

Line 97-107 CURB-65 Severity Score <1 should be less and equal; the norm of mentioned lab test are not so obvious or a bit different from laboratory to laboratory, consider to add normal range

Authors’ reply: We really appreciate this comment in order to improve our paper. We have added these recommendations (line 101-112).

Line 112-115- patient had the possibility of being treated – re-phrase

Authors’ reply: We really appreciate this comment in order to improve our paper. We have re-phrased it (“Patients could be treated…”) (line 136)

Line 152- add reference.

Authors’ reply: We really appreciate this comment in order to improve our paper. We have re-phrased this sentence.

Line 157 –add reference (e.g Teijaro JR. Cytokine storms in infectious diseases. Semin Immunopathol. 2017;39(5):501-503. doi:10.1007/s00281-017-0640-2)

Authors’ reply: We really appreciate this comment in order to improve our paper. We have added this reference.

Line 156-157 was those lab test mandatory according to clinical protocols? It is not clear indication for immunomodulatory agents to be used, which were the first choice - steroids? Other immunomodulatory medication.

Authors’ reply: We really appreciate this comment in order to improve our paper. Those laboratory tests were recommended in the clinical protocols. The choice of which therapy to use was made by physicians according to their own clinical judgment. We have added it (line 182).

Line 186 –‘severe’ or prolonged?

Authors’ reply: We really appreciate this comment in order to improve our paper. It is “severe COVID-19…”.

Discussion

Line 210- ‘and respiratory and kidney failure ’ change to multiorgan failure.

Authors’ reply: We really appreciate this comment in order to improve our paper. We have changed it.

Line 239- the severest – the most severe.

Authors’ reply: We really appreciate this comment in order to improve our paper. We have changed it.

Line 240-241 – ‘Thus, more complex and longer antiviral therapies were recommended according to the severity of COVID-19 in our protocols.’ – What the information authors would like to pass to the readers? Knowing that there is no solid evidence for any therapy to be true effective, about what, authors informed us couple of times through this manuscript

Authors’ reply: We really appreciate this comment in order to improve our paper. We would like to show that there is no solid evidence for any therapy to be effective.

Line 285 – what kind of type one-off recommendation of adjunctive therapies authors are mentioned?

Authors’ reply: We really appreciate this comment in order to improve our paper. We have delated this sentence because there was no specific recommendations of other adjunctive therapies in the protocols.

Line 289-303 – is repetition of what is described in paragraph 3

Authors’ reply: We really appreciate this comment in order to improve our paper. We have adapted this paragraph.

References

Review references and upgrade as most of them were published in April ‘20

Authors’ reply: We really appreciate this comment in order to improve our paper. We have reviewed and updated the references.

Tables

Table 2.

In last column – severe pneumonia - sum up numbers give 14 not 15, what regime is missing?

Authors’ reply: We really appreciate this comment in order to improve our paper. Sum up numbers give 15.

Table 1 and 3 – use bold to underline agents

Authors’ reply: We really appreciate this comment in order to improve our paper. We have used bold to underline agents.

Reviewer 2 Report

This study showed that the COVID-19 protocols of the largest hospitals in
Andalusia, Spain, varied widely in several aspects, including antiviral treatment; corticosteroids.

It is well designed and the results well showed.

Minor points

Conclusion

lines 338-339 Please clarify: There was greater  consensus regarding the use of tocilizumab; prophylactic heparin for all patients,

Please clarify if tocilizumab might reduce the risks, for example reduce the risk ofinvasive mechanical ventilation or death in patients with severe COVID-19 pneumonia, etc.

Author Response

Lyn Zhao

Assitant Editor

International Journal of Environment Research and Public Health

September 14, 2020

Dear Dr. Lyn Zhao:

Thank you for reviewing and considering our manuscript, “Differences between clinical protocols for the management of coronavirus disease 2019 (COVID-19) in Andalusia, Spain” for possible publication in the International Journal of Environment Research and Public Health. We greatly appreciate the comments provided to us by the reviewers.

We have made revisions to our manuscript, according the recommendations of the reviewers, and are now resubmitting it to you for your consideration. The reviewers’ comments and our corresponding responses are listed below.

Thank you very much for your kind attention on this matter.

Sincerely

Corresponding authors on behalf of all co-authors.

Luis M. Pérez-Belmonte, MD, PhD

María D. López-Carmona, MD, PhD

luismiguelpb1984@gmail.com; mdlcorreo@gmail.com

Reviewer #2:

This study showed that the COVID-19 protocols of the largest hospitals in
Andalusia, Spain, varied widely in several aspects, including antiviral treatment; corticosteroids.

It is well designed and the results well showed.

Authors’ reply: We really appreciate this comment.

Minor points

Conclusion

lines 338-339 Please clarify: There was greater  consensus regarding the use of tocilizumab; prophylactic heparin for all patients.

Authors’ reply: We really appreciate this comment in order to improve our paper. In order to clarify this issue, we have changed the sentence “There was greater consensus regarding the use of tocilizumab; prophylactic heparin for all patients, therapeutic heparin for evidence of venous thromboembolism, general symptomatic and supportive care, and hospital organization. for “The use of tocilizumab for selected patients; prophylactic heparin for all patients, therapeutic heparin for evidence of venous thromboembolism, general symptomatic and supportive care, and hospital organization were recommended in all protocols”.

Please clarify if tocilizumab might reduce the risks, for example reduce the risk of invasive mechanical ventilation or death in patients with severe COVID-19 pneumonia, etc.

Authors’ reply: We really appreciate this comment in order to improve our paper. About tocilizumab we have written in the discussion: “Tocilizumab, an anti-interleukin 6 receptor antibody, has been used in small studies with early reports of success. A dose of 400mg was associated with clinical improvements and successful discharge, with most patients only receiving a single dose [27,28]. Several randomized clinical trials on tocilizumab, either alone or in combination, in patients with COVID-19 with severe pneumonia are underway and its use was included in the Chinese national treatment guidelines [28,29]. Likewise, tocilizumab was recommended in all protocols in Andalusian hospitals; the protocols indicate a single dose of 400mg or 600mg, according to body weight, and a second dose 12-24 h later, if required. Although tocilizumab is a promising therapy, the data currently available are still too limited to draw any conclusion about its viability.”

Reviewer 3 Report

I read with great interesting this manuscript. I find it well wrote and with good idea research.

Below my suggestions:

  1. Introduction: please update burden of COVID wordwilde and in Spain around data of revision and the trend from Febraury to September
  2. Methods: well wrote
  3. Results:no suggestion. 
  4. Discussion: about use of antiviral add in discussion this interesting results from Italy (COVID-19 RISK and Treatments (CORIST) Collaboration members:. Use of hydroxychloroquine in hospitalised COVID-19 patients is associated with reduced mortality: Findings from the observational multicentre Italian CORIST study [published online ahead of print, 2020 Aug 25]. Eur J Intern Med. 2020;S0953-6205(20)30335-6. doi:10.1016/j.ejim.2020.08.019). Add better future perspectives from your study (see and cite Di Gennaro F, Pizzol D, Marotta C, et al. Coronavirus Diseases (COVID-19) Current Status and Future Perspectives: A Narrative Review. Int J Environ Res Public Health. 2020;17(8):2690. Published 2020 Apr 14. doi:10.3390/ijerph17082690)
  5. Also Discuss better the role of risk factors for hospital mortality (Di Castelnuovo, A et al Common cardiovascular risk factors and in-hospital mortality in 3,894
    317 patients with COVID-19: survival analysis and machine learning-based findings from the multicentre
    318 Italian CORIST Study. Nutrition, Metabolism and Cardiovascular Diseases. 2020. Jul 31.
    319 https://doi.org/10.1016/j.numecd.2020.07.031.)
  6. Conclusion: are well written and coherent with the results

Author Response

Lyn Zhao

Assitant Editor

International Journal of Environment Research and Public Health

September 14, 2020

Dear Dr. Lyn Zhao:

Thank you for reviewing and considering our manuscript, “Differences between clinical protocols for the management of coronavirus disease 2019 (COVID-19) in Andalusia, Spain” for possible publication in the International Journal of Environment Research and Public Health. We greatly appreciate the comments provided to us by the reviewers.

We have made revisions to our manuscript, according the recommendations of the reviewers, and are now resubmitting it to you for your consideration. The reviewers’ comments and our corresponding responses are listed below.

Thank you very much for your kind attention on this matter.

Sincerely

Corresponding authors on behalf of all co-authors.

Luis M. Pérez-Belmonte, MD, PhD

María D. López-Carmona, MD, PhD

luismiguelpb1984@gmail.com; mdlcorreo@gmail.com

Reviewer #3:

I read with great interesting this manuscript. I find it well wrote and with good idea research.

Authors’ reply: We really appreciate this comment.

Below my suggestions:

  1. Introduction: please update burden of COVID wordwilde and in Spain around data of revision and the trend from Febraury to September.

Authors’ reply: We have updated numbers of COVID-19 in Spain and Andalusia.

  1. Methods: well wrote

Authors’ reply: Thank you for your comment.

  1. Results: no suggestion. 

Authors’ reply: Thank you for your comment.

  1. Discussion: about use of antiviral add in discussion this interesting results from Italy (COVID-19 RISK and Treatments (CORIST) Collaboration members:. Use of hydroxychloroquine in hospitalised COVID-19 patients is associated with reduced mortality: Findings from the observational multicentre Italian CORIST study [published online ahead of print, 2020 Aug 25]. Eur J Intern Med. 2020;S0953-6205(20)30335-6. doi:10.1016/j.ejim.2020.08.019). Add better future perspectives from your study (see and cite Di Gennaro F, Pizzol D, Marotta C, et al. Coronavirus Diseases (COVID-19) Current Status and Future Perspectives: A Narrative Review. Int J Environ Res Public Health. 2020;17(8):2690. Published 2020 Apr 14. doi:10.3390/ijerph17082690).

Authors’ reply: We really appreciate this comment in order to improve our paper. We have discussed these issues in this section of the paper and have included the references (lines 248-250 and lines 350-353).

  1. Also Discuss better the role of risk factors for hospital mortality (Di Castelnuovo, A et al Common cardiovascular risk factors and in-hospital mortality in 3,894
    317 patients with COVID-19: survival analysis and machine learning-based findings from the multicentre 318 Italian CORIST Study. Nutrition, Metabolism and Cardiovascular Diseases. 2020. Jul 31. 319 https://doi.org/10.1016/j.numecd.2020.07.031.).

Authors’ reply: We really appreciate this comment in order to improve our paper. We have discussed this issue in this section of the paper and have included the reference (line 235-237)

  1. Conclusion: are well written and coherent with the results.

Authors’ reply: Thank you for your comment.

This manuscript is a resubmission of an earlier submission. The following is a list of the peer review reports and author responses from that submission.